# Implementation fidelity of Ethiopia's Malaria test-and-treat guideline amid a resurgence in Amhara Region: A mixed-methods study

Mastewal Worku Lake[1,2]*, Kassahun Alemu Gelaye[3], Mulusew Andualem Asemahagn[4], Kindie Fentahun Muchie[1], Hailemariam Awoke Engedaw[5], Muluken Azage Yenesew[6]

1 Department of Epidemiology and Biostatistics, School of Public Health, College of Medicine and Health Sciences, Bahir Dar University, Bahir Dar, Ethiopia, 2 Amhara Public Health Institute, Bahir Dar, Ethiopia, 3 Department of Epidemiology and Biostatistics, Institute of Public Health, University of Gondar, Gondar, Ethiopia, 4 Department of Health Systems Management and Health Economics, School of Public Health, College of Medicine and Health Sciences, Bahir Dar University, Bahir Dar, Ethiopia, 5 Department of Internal medicine, School of Medicine, College of Medicine and Health Sciences, Bahir Dar University, Bahir Dar, Ethiopia, 6 Department of Environmental Health, School of Public Health, Bahir Dar University, Bahir Dar, Ethiopia

* matewal.worku@gmail.com

## Abstract

### Background

Ethiopia has experienced a marked malaria resurgence in recent years, with the Amhara Region disproportionately affected. Although Ethiopia's national strategy emphasizes test-before-treat and a public–private mix, implementation fidelity of the malaria test-and-treat guideline during resurgence has not been well characterized. This study assessed fidelity to malaria diagnosis and treatment guidelines in public and private health facilities in the Amhara Region within this resurgence context.

### Methods

We conducted a convergent parallel mixed-methods study from February to March 2025 in 53 health facilities (38 public, 15 private) in Amhara Region. The facility was the unit of analysis; one provider primarily responsible for malaria case management was interviewed per facility (n = 53). Implementation fidelity was operationalized using Carroll's framework across three domains: content (adherence to key diagnostic/treatment steps), coverage (proportion of suspected cases tested before treatment, extracted from facility registers), and frequency (self-reported consistency of testing for febrile patients in the preceding month). Domain scores were standardized to 0–100 and averaged with equal weights to form a composite fidelity score; ≥ 75% indicated high fidelity. To explain quantitative patterns, we conducted 32 in-depth interviews and analyzed data using inductive thematic analysis with CFIR-informed interpretation. Quantitative analysis used nonparametric tests and parsimonious

**Data availability statement:** Data cannot be shared publicly due to containing potentially identifying information. Data access is restricted by the Bahir Dar University College of Medicine and Health Sciences Institutional Review Board (BDU-CMHS IRB). De-identified quantitative data, codebooks, and analysis code are available upon request to the IRB (Ref: 3053/2024) (cmhs@bdu.edu.et). Selected de-identified qualitative excerpts are available to interested researchers with IRB approval.

**Funding:** The author(s) received no specific funding for this work.

**Competing interests:** The authors have declared that no competing interests exist.

multivariable linear regression, with prespecified sensitivity analyses excluding the self-reported frequency domain.

## Results

Overall mean implementation fidelity was 64.3% (SD 12.1); 40% of facilities had high fidelity (≥75%), and 13% scored <50%. Public facilities had higher fidelity than private facilities (median 67% [IQR 60–77] vs 63% [IQR 56–70]; Wilcoxon rank-sum $p = 0.041$). In multivariable analysis, higher fidelity was associated with higher participant responsiveness (β = 3.4, $p < 0.001$), stronger facilitation strategies (β = 2.8, $p < 0.001$), and lower perceived intervention complexity (reverse-coded; β = 2.1, $p < 0.001$). Interviews indicated that fidelity gaps were driven by diagnostic and treatment deviations (including non–species-specific prescribing), inconsistent counseling and follow-up mechanisms, supply constraints, and patient pressure, with challenges more frequently emphasized in private facilities.

## Conclusions

Implementation fidelity to Ethiopia's malaria test-and-treat guideline in Amhara during resurgence was moderate, with lower fidelity in private facilities. Provider responsiveness, facilitation strategies, and lower intervention complexity were identified as factors associated with implementation fidelity. Strengthening supportive supervision and mentorship with explicit inclusion of private facilities, improving supply reliability, and simplifying decision supports may improve adherence during resurgence.

## Introduction

Malaria is an infectious disease caused by the Plasmodium parasites [1]. It is an entirely preventable and treatable parasitic disease [2,3]. Globally, an estimated 263 million cases and 597,000 deaths occurred in 2023, with 94% of deaths occurring in sub-Saharan [4,5]. Factors contributing to this resurgence include insecticide resistance, the spread of Anopheles stephensi, climate anomalies, conflict, healthcare system disruptions, and a resources shortages [4,6]. Alarmingly, molecular markers associated with partial artemisinin resistance (e.g., pfk13 R561H, P574L, A675V) have been reported in multiple African countries, including Ethiopia [7].

Despite historical gains, Ethiopia has faced a significant malaria resurgence, with confirmed cases rising from 1.0 million in 2018 to 4.1 million in 2023 [5], and exceeding 8.3 million by the third quarter of 2024 [8]. The Amhara Region bears a disproportionate burden, contributing approximately 33% of Ethiopia's cases [9,10]. This resurgence stems from intersecting factors: climate anomalies [5,11], health system fragility exacerbated by conflict, affecting up to 60% of services [12], inconsistent implementation of the national "test-before-treat" strategy [13–15], the expansion of Anopheles stephensi into urban settings [16] and increasing proportions of P. vivax in border areas [17,18].

Ethiopia adopted a national test-before-treat strategy in 2017, mandating parasitological confirmation before treatment. However, implementation has been inconsistent. In 2021, only 58% of suspected cases were tested nationally, and the private sector frequently experiences shortages of diagnostic tests and antimalarial medicines [19,20]. To expand access, Ethiopia utilizes a Public-Private Mix (PPM) strategy, with the private sector now managing approximately 40% of malaria-related visits in urban and border areas [21]. However, prior evidence suggests that adherence to diagnostic protocols in private clinics may be as low as 21% [20,22]. Our study examines whether PPM structures in the Amhara region reduce such dissipations and close these gaps, providing new insights into scalable private-sector engagement during resurgence.

While malaria is a preventable and curable disease [1], the effectiveness of national 'test-and-treat' strategies depends heavily on implementation fidelity, the degree to which the guideline is delivered as intended by its developers [23]. Adaptive fidelity refers to protocol modifications that maintain core intervention principles while adjusting to contextual constraints. Studies in non-malaria contexts in Ethiopia and Kenya suggest that high fidelity, supported by supervision, consistent supply chains, and provider engagement, improves outcomes even in resource-limited settings [24]. In malaria-specific contexts, evidence from Nigeria shows that participant responsiveness, availability of guidelines, and diagnostic tools strongly predict fidelity, with public facilities typically outperforming private ones [25]. Despite Ethiopia's policy efforts and PPM leverage, malaria cases have surged, particularly in the Amhara Region [8,10,26], highlighting weak guideline implementation, especially in private facilities, and a critical evidence gap regarding fidelity amidst health system disruptions [22].

Prior studies in Ethiopia have largely focused on malaria outcomes or individual clinician practices rather than systematically measuring implementation fidelity using established frameworks across both public and private facilities in resurgence-affected contexts [22,27]. Therefore, this study aimed to assess implementation fidelity of Ethiopia's malaria test-and-treat guideline in public and private health facilities in the Amhara Region during the resurgence, and to explain quantitative fidelity patterns using qualitative evidence on implementation determinants. We applied Carroll's implementation fidelity framework to quantify the fidelity domains of content, coverage, and frequency (23), and we used the Consolidated Framework for Implementation Research to characterize multilevel determinants of implementation [28]. Findings are intended to inform targeted strategies aligned with Ethiopia's malaria strategy and to provide practical lessons for improving guideline implementation in conflict-affected, resource-constrained settings [29].

## Methods and materials

### Study design

We used a convergent parallel mixed-methods design [30], comprising (i) a facility-based cross-sectional assessment to quantify implementation fidelity and (ii) in-depth interviews to explain quantitative findings by identifying multilevel determinants of implementation. Quantitative and qualitative data were collected during the same period, analyzed separately, and then integrated using joint displays [31].

### Conceptual framework

We evaluated fidelity using Carroll's implementation fidelity framework [23], focusing on three domains: content (adherence to diagnostic and treatment steps), coverage (proportion of suspected malaria cases tested before treatment), and frequency (consistency of testing suspected cases before treatment). Moderating factors included facilitation strategies, intervention complexity, and participant responsiveness, interpreted using the Consolidated Framework for Implementation Research (CFIR) [28] (Fig 1).

Conceptual framework illustrating the fidelity of implementation (FOI) of the malaria test-and-treat strategy in selected public and private health facilities in the Amhara Region, Ethiopia (2025). The framework highlights key components,

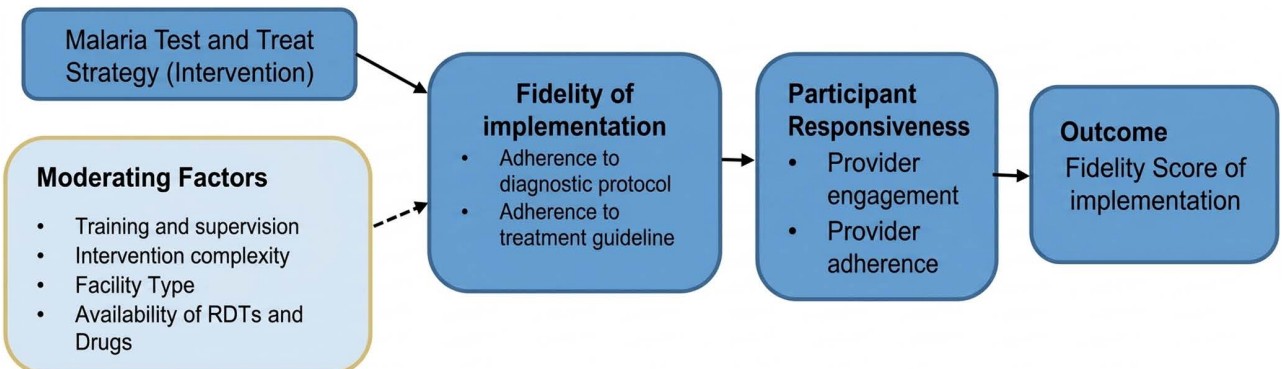

**Fig 1. Conceptual framework for fidelity of implementation of the malaria test-and-treat strategy.**

including fidelity (adherence to diagnostic and treatment protocols), participant responsiveness (healthcare provider adherence to guidelines), and potential moderating factors, such as facilitation strategies, intervention complexity, and participant engagement.

## Study settings and period

We conducted the study from 1 February to 30 March 2025 in the Amhara Region, Ethiopia (9°20'–14°20' N, 36°20'–40°20' E). The region comprises 22 administrative zones (14 zones, 8 city administrations), 236 Woredas (districts), and 4,060 kebeles, with an estimated population of 25 million [ 32]. Of the 236 woredas in the Amhara Region, 166 had complete weekly malaria surveillance data and were eligible for resurgence screening.

Malaria transmission occurs across the region, and malaria services are delivered through public and private facilities, including hospitals, health centers, and health posts. Public and private health facilities providing malaria diagnosis and treatment were included. *19 woredas were selected via simple random sampling, stratified by geographic zone to ensure regional representation* (Fig 2).

## Resurgence operational definition and sampling frame

Facilities were sampled to represent both the public and private sectors and variation in facility type and geographic context. Woredas were prioritized based on documented malaria resurgence. Using weekly surveillance data from the Amhara Public Health Institute, we defined resurgence as a 50% or greater increase in confirmed malaria cases in 2024 compared with 2022. Of the 236 woredas in Amhara Region, 166 had complete weekly malaria surveillance data and were eligible for resurgence screening; 30 met the resurgence criterion. From those 30 woredas, 19 were selected by simple random sampling, stratified by geographic zone to ensure regional representation. Public health facilities were selected using simple random sampling from regional master lists. Private facilities were selected using a two-stage approach because a complete regional private-facility registry was not available: (1) woredas were selected, and (2) private facilities within selected woredas were sampled systematically from locally available lists maintained by woreda health offices. Sampling and participation are summarized in S1 Fig.

## Participants and unit of analysis

The facility was the unit of analysis. In each facility, one healthcare provider primarily responsible for malaria case management was selected as the facility-level key informant if they had at least 6 months of experience in malaria diagnosis

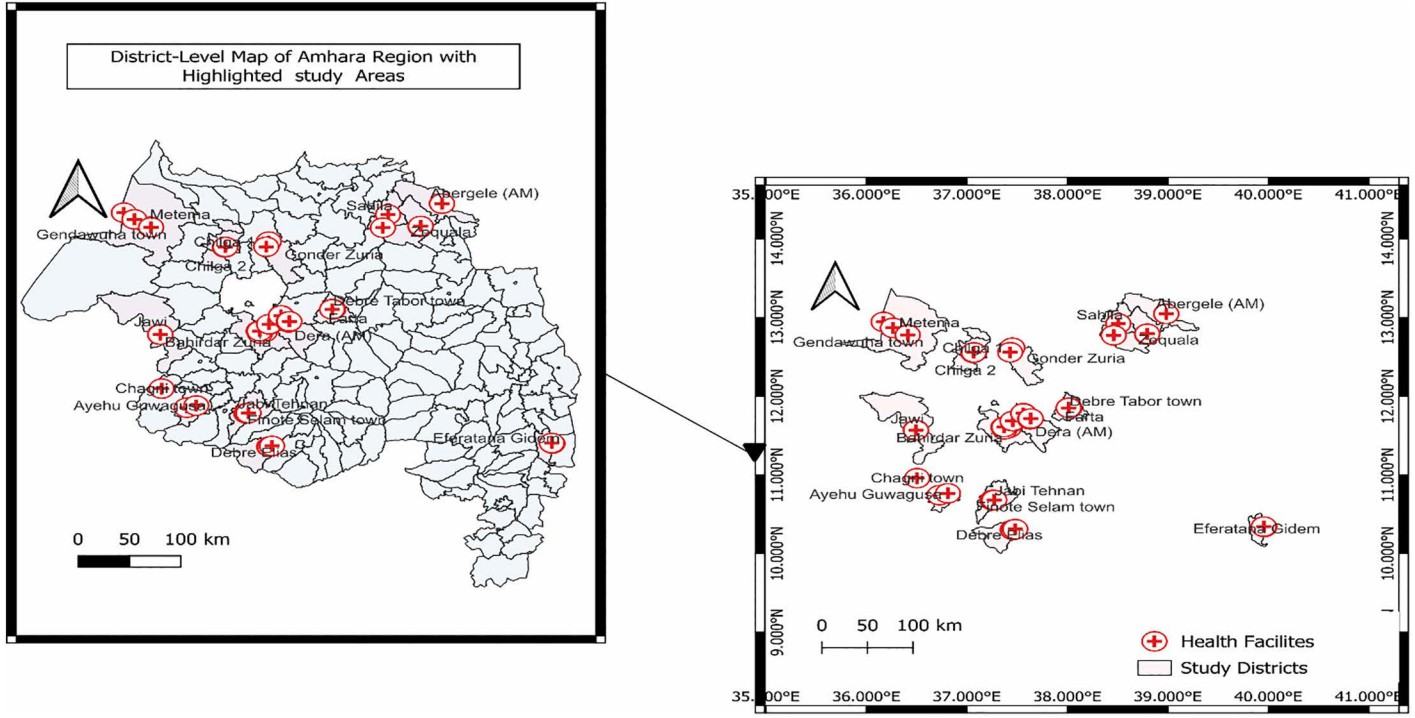

**Fig 2. Map of Amhara Regions shows the study woredas.** The map was produced by the authors using QGIS software and publicly available administrative boundary layers from the Ethiopian Central Statistical Agency and Natural Earth.

and/or treatment. Providers who were on leave during data collection were excluded. This approach provided a single facility-level report on implementation processes and supported aggregation of facility-level covariates, while register review was used to triangulate selected indicators [23,33].

## Sample Size

The sample size was determined based on the primary outcome, implementation fidelity to national malaria guidelines. In the absence of prior studies quantifying facility-level fidelity using a validated scale in this setting, fidelity was initially operationalized as a binary variable (good vs poor), and the required sample size for a single proportion was estimated using the standard formula [34, 35]:

$$n = Z^2p(1 - p/d^2).$$

Assuming a conservative prevalence of good fidelity of 50%, a two-sided significance level of 5% (Z = 1.96), and a margin of error of 0.13, the estimated minimum sample size was 57 facilities. Due to feasibility constraints and the number of eligible facilities within the selected woredas, the final sample included 53 health facilities.

In addition, guided by the CFIR [28], we considered intervention complexity as a potential moderator of fidelity. Assuming that approximately 60% of facilities would perceive the guideline as complex, a sample of 53 facilities provided approximately 80% power to detect a 20 percentage-point difference in fidelity between facilities perceiving the guideline as complex and those that did not.

## Qualitative sampling

For the qualitative component, we used maximum variation sampling to capture diversity in provider cadre, facility type (public/private), and malaria case-management experience, consistent with the CFIR domain "Characteristics of Individuals" [28]. In-depth interviews continued until thematic saturation was reached. A total of **32 providers** participated.

## Eligibility criteria

**Inclusion criteria.** Healthcare professionals with at least 6 months of experience in malaria diagnosis and treatment, and currently employed at facilities providing these services.

**Exclusion criteria.** Healthcare professionals with less than 6 months of relevant experience, those on leave during the study period, and those employed at facilities that did not provide malaria diagnosis and treatment services.

## Variables and measures

Fidelity was computed as an unweighted average of three standardized domain scores (0–100): content, coverage, and frequency.

Content (adherence) captured completion of four key guideline components at the facility: (i) parasitological testing for suspected cases, (ii) species-specific treatment selection, (iii) observed first dose at the facility when feasible, and (iv) provision of key counseling messages. Items were obtained primarily from provider report; species-specific treatment was additionally cross-checked using targeted review of recent malaria register entries and/or prescriptions available during the visit (S1 Table). Each item was scored 0/1 (sum 0–4) and standardized to 0–100.

Coverage (register-extracted) was defined as the percentage of suspected malaria cases tested before treatment, extracted from facility registers for the most recent routine reporting period available (S1 Table).

Frequency (self-reported) was defined using a single Likert-type item on how often providers ordered malaria tests for suspected malaria or febrile patients before treatment in the preceding month (never, rarely, sometimes, often, always), transformed linearly to a 0–100 scale. Because this domain relied on self-report and was potentially susceptible to social desirability bias, we prespecified sensitivity analyses excluding the frequency domain [36,37], we prespecified sensitivity analyses excluding the frequency domain.

## Standardization and composite scoring

Each fidelity construct was standardized to a 0–100 scale using the formula

$$\text{Observed score/ maximum possible score} \times 100$$

The composite fidelity score was calculated as the unweighted mean of the standardized content, coverage, and frequency scores. Equal weighting was prespecified based on Carroll's framework and the WHO test-and-treat strategy, which emphasizes diagnostic access, clinical adherence, and sustained practice as core domains of implementation. Fidelity can be high (≥75%) [33], medium (50–74%), or low (<50%). A 50% threshold is often used to define low fidelity, suggesting minimal program impact [38,39].

## Covariates

Resurgence-related covariates included facility-level stockout duration for first-line malaria diagnostics or antimalarial medicines in the preceding 3 months and district-level percentage change in confirmed malaria cases between 2022 and 2024. Facility characteristics included sector (public/private), facility level/type, guideline availability, and diagnostic modality (microscopy, RDT, or both). Provider characteristics included sex, age group, profession, and years of experience.

### Implementation moderators (CFIR-aligned)

The main implementation moderators were:

Facilitation strategies: training, supervision, mentorship, job aids, and availability of diagnostics and medicines;

Participant responsiveness: provider acceptance of and engagement with the guideline;

Intervention complexity: perceived difficulty of implementing the malaria test-and-treat guideline.

### Data collection

Fidelity and implementation moderators were assessed using a structured 56-item questionnaire implemented in Kobo Toolbox and adapted from published literature and implementation fidelity frameworks. The full questionnaire is provided in S1 File, and the mapping of analyzed variables to questionnaire items is summarized in S1 Table. The tool was pilot tested before data collection, and data collectors received two days of training.

For the composite fidelity outcome, the frequency domain was derived from a single questionnaire item assessing how often suspected malaria cases were tested before treatment in the preceding month. Other questionnaire items relating to refresher training, supervision, routine assessment of supplies, and quality-assurance activities were not included in the frequency domain score; instead, they were used descriptively and/or to inform facilitation and contextual variables.

For the qualitative component, we used a semi-structured interview guide adapted from the literature, pretested before data collection, and designed to explore barriers and facilitators to implementation (S2 Table). Trained interviewers conducted in-depth interviews in Amharic using semi-structured conversational prompts. Interviews lasted approximately 40–60 minutes, were audio-recorded with permission, transcribed verbatim, and translated into English. Daily debriefings and transcript reviews were used to maintain data quality and consistency. Qualitative data were managed and analyzed using ATLAS.ti version 23 (ATLAS.ti Scientific Software Development GmbH, Berlin, Germany). Following translation and transcription verification, all 32 transcripts were imported into ATLAS.ti as primary documents. All coding, memo writing, network visualization of code relationships, and theme development activities were performed within the platform to ensure a systematic, transparent, and auditable analytical trail, consistent with standards for rigorous qualitative data management.

## Data analysis

### Quantitative analysis

Facility and provider characteristics were summarized using means with standard deviations (SD), medians with inter-quartile ranges (IQR), and proportions. Because the fidelity score was not normally distributed (Shapiro-Wilk test), group comparisons used Mann-Whitney U tests for two-group comparisons and Kruskal-Wallis tests for comparisons involving three or more groups [40]. Categorical associations were assessed using Fisher's exact test where appropriate.

Internal consistency of multi-item scales was evaluated using Cronbach's alpha [41]. The frequency domain was based on a single item and therefore, was not included in internal consistency testing. To identify factors associated with fidelity, we fitted parsimonious hierarchical multivariable linear regression models adjusting for resurgence-related factors, implementation moderators, and facility and provider characteristics. Because the outcome was bounded from 0 to 100, model diagnostics were examined carefully, including residual plots and normality of residuals. Fractional regression models were explored as sensitivity analyses after rescaling the outcome, and these produced substantively similar findings; therefore, linear regression results are presented for interpretability. Statistical significance was set at **$p < 0.05$**. All analyses were conducted in **R version 4.5.1** [42].

The relationship was modeled as:

$$Y_i = \beta_0 + \beta_1 Xi_1 + \beta_2 Xi_2 + ... + \beta_n Xi_n + \varepsilon_i$$

where Yi is the fidelity score, $\beta_0$ is the intercept, Xi are the predictors, $\beta_n$ are regression coefficients, and $\varepsilon_i$ is the error term.

We fitted two hierarchical models. Model 1 included resurgence-related and contextual covariates. Model 2 added implementation moderators, including participant responsiveness, facilitation strategies, and intervention complexity. Multi-collinearity was assessed using variance inflation factors, with all values <2.0.

### Sensitivity analysis

To address potential social desirability bias in the self-reported frequency domain, we compared the primary composite fidelity score (equal-weighted content, coverage, and frequency) with an alternative score excluding the single self-reported frequency item. Mann-Whitney U tests were repeated to assess whether the public-private difference remained consistent across scoring methods.

### Qualitative analysis

We conducted inductive thematic analysis to generate codes and themes from the interview data. After themes were finalized, we mapped them to CFIR domains and constructs to support interpretation and mixed-method integration. This approach allowed the analysis to remain data-driven while using CFIR to organize explanations of implementation determinants. Trustworthiness was supported using Guba's criteria [43], including credibility through triangulation and prolonged engagement, transferability through rich description and illustrative quotations, dependability through independent coding and investigator consensus, and confirmability through triangulation of interview findings with medical records [44] The coding framework and CFIR mapping are summarized in Table 1, and the interview guide is provided in S2 Table.

### Mixed-methods integration

Quantitative and qualitative data were collected concurrently and analyzed independently, then integrated using a joint display matrix [31]. This approach aligned interview themes with key quantitative variables, including participant responsiveness, facilitation strategies, and intervention complexity, allowing findings to be compared for convergence, complementarity, or divergence.

### Ethical approval

The study received ethical approval from the Bahir Dar University College of Medicine and Health Sciences Institutional Review Board (BDU-CMHS-IRB Ref: 3053/2024). All participants provided written informed consent before data collection. Participants were informed about the study purpose, procedures, risks, and benefits in a language they understood. Interviews were conducted in private settings to maintain confidentiality. Audio files were stored on password-protected devices and deleted after transcription. Transcripts and quantitative datasets were de-identified prior to analysis, and participants could withdraw at any time without penalty. No financial compensation was provided.

## Results

### Participant and facility characteristics

We assessed 53 facilities (38 public, 15 private) and interviewed one primary malaria case-management provider per facility (n = 53). The workforce was predominantly male (81%), with a higher proportion of males in private facilities (87%). The average service length was 9.8 years (SD = 5.9), with public-sector healthcare providers having longer tenures than those in the private sector (10.7 vs. 7.5 years). Most healthcare providers were aged 30–39, suggesting an experienced mid-career group. Nurses comprised the largest professional group (45%), while medical doctors were more prevalent in

**Table 1. CFIR Construct Mapping for Qualitative Coding.**

| Qualitative theme/ subtheme | Description of what was observed | CFIR domain | CFIR construct(s) | Relevance to implementation fidelity |
|---|---|---|---|---|
| Microscopy adherence | Most providers reported routine microscopy use for suspected malaria cases where available | Inner setting | Structural characteristics; Available resources | Supports diagnostic fidelity when supplies and trained staff are available |
| Use of multispecies RDTs as an adaptive response | Some providers preferred multispecies RDTs when microscopy reagents were unavailable or considered unreliable | Intervention characteristics; Inner setting | Adaptability; Available resources | Reflects adaptive implementation under resource constraints |
| Presumptive/clinical diagnosis during shortages or high workload | Providers reported occasional treatment without confirmatory testing during stockouts, severe illness, or patient pressure | Inner setting; Outer setting | Available resources; Structural characteristics; Patient needs and resources | Indicates drift from test-before-treat fidelity |
| Non–species-specific treatment selection | Some providers reported using non–species-specific regimens because of diagnostic uncertainty, medicine shortages, or concerns about laboratory quality | Characteristics of individuals; Inner setting | Knowledge and beliefs about the intervention; Available resources | Reduces adherence to treatment content of the guideline |
| Non-standard antimalarial use in some private facilities | A few private providers described use of non-standard artemether-lumefantrine brands, often linked to affordability and business pressures | Outer setting; Characteristics of individuals; Inner setting | External policy and incentives; Knowledge and beliefs about the intervention; Implementation climate | Suggests erosion of treatment fidelity, especially in private settings |
| First-dose observation not routinely implemented | Providers in both sectors reported that most patients took the first dose at home rather than under observation | Inner setting; Process | Compatibility; Available resources; Executing | Weakens fidelity to recommended treatment delivery procedures |
| Limited counseling messages | Counseling on treatment completion, danger signs, and follow-up was often incomplete or inconsistent | Characteristics of individuals; Inner setting | Knowledge and beliefs about the intervention; Access to knowledge and information | Reduces fidelity to patient communication components of care |
| Weak or absent follow-up mechanisms | Providers reported no formal mechanism for follow-up after treatment initiation in most facilities | Process; Inner setting | Reflecting and evaluating; Networks and communications | Limits continuity of care and reinforcement of treatment adherence |
| Limited patient involvement in diagnosis and treatment decisions | Providers reported that diagnosis and treatment plans were not always discussed with patients, especially when tests were negative but symptoms persisted | Outer setting; Process | Patient needs and resources; Engaging | May contribute to poor adherence, pressure for inappropriate treatment, and reduced fidelity |

**Note:** Themes were generated inductively from interview data and subsequently mapped to CFIR domains and constructs to support interpretation.

private facilities (33% vs. 11%). Access to national malaria guidelines differed significantly between sectors; 82% of public facilities had on-site copies compared to 33% of private facilities (p < 0.001) (Table 2).

## Diagnostic practices and supply interruptions

Public and private health facilities employed significantly different malaria testing algorithms (p < 0.001). Over half of public facilities (55%) employed both light microscopy and rapid diagnostic tests (RDTs), whereas 45% relied solely on microscopy. In contrast, 80% of private facilities used microscopy alone, and 20% used RDTs alone; none combined the two methods. When aggregated, microscopy alone was the dominant approach (55%), followed by dual testing (40%) and RDTs alone (5.7%) (Table 3). Stockouts of Giemsa stain or microscope slides were reported in 41% of facilities.

## Healthcare provider knowledge and reported treatment practices

Healthcare providers excelled in microscopy knowledge (98.1%), artesunate use (98.1%), and first-line treatment (98.1%), but struggled with primaquine use (28.3% correct) and RDT Reliability: 20.8% correct interpreted RDT limitations. Private facilities reported occasional use of non-standard AL brands (11%), citing client affordability (Fig 3).

**Table 2. Characteristics of Healthcare Facilities and Healthcare Providers, Amhara Region 2025 (N = 53).**

| Characteristic | Overall (N = 53) | Private (n = 15) | Public (n = 38) |
|---|---|---|---|
| Provider sex (male) | 43 (81.1%) | 13 (86.7%) | 30 (78.9%) |
| Provider age group (years) | | | |
| <30 | 10 (18.9%) | 5 (33.3%) | 5 (13.2%) |
| 30–39 | 36 (67.9%) | 8 (53.3%) | 28 (73.7%) |
| 40–49 | 6 (11.3%) | 2 (13.3%) | 4 (10.5%) |
| ≥50 | 1 (1.9%) | 0 (0.0%) | 1 (2.6%) |
| Profession | | | |
| Medical doctor | 9 (17.0%) | 5 (33.3%) | 4 (10.5%) |
| Nurse | 24 (45.3%) | 6 (40.0%) | 18 (47.4%) |
| Health officer/other | 20 (37.7%) | 4 (26.7%) | 16 (42.1%) |
| National malaria guideline available on-site | 36 (67.9%) | 5 (33.3%) | 31 (81.6%) |
| Years of service, mean (SD) | 9.8 (5.9) | 7.5 (5.2) | 10.7 (6.0) |

Notes: Values are n (%) unless otherwise indicated.

**Table 3. Association Between Facility Type and Malaria Test Method, Amhara Region, 2025.**

| | Test Method | | | Total | p-value[1] |
|---|---|---|---|---|---|
| | **Microscope** | **RDT** | **Both** | | |
| Facility Type | | | | | <0.001 |
| Public | 17 (45%) | 0 (0%) | 21 (55%) | 38 (100%) | |
| Private | 12 (80%) | 3 (20%) | 0 (0%) | 15 (100%) | |
| Total | 29 (55%) | 3 (5.7%) | 21 (40%) | 53 (100%) | |

[1] Fisher's exact test

## Facilitation strategies

The implementation of facilitation measures varied. Job aids were the most common support (40% of facilities), followed by available diagnostic supplies/tools (32%). Less frequent were human-resource-intensive supports: supervision (17%), training (13%), and mentorship (11%). This aligns with mixed-methods findings indicating a preference for material resources over sustained capacity building (Fig 4).

## Fidelity to National Malaria "Test and Treat" Protocol

The mean facility-level fidelity score was 64.3% (SD = 12.1). Forty percent of facilities achieved high fidelity (≥75%), 47% medium fidelity (50–74%), and 13% low fidelity (<50%). Median fidelity was significantly higher in public facilities (67% [IQR 60–77]) than in private facilities (63% [IQR 56–70]) (Wilcoxon rank-sum (Mann–Whitney U) = 231, p = 0.041).. Only 51% of facilities showed evidence of species-specific prescribing, and 11% of private facilities reported occasional use of non-standard artemether-lumefantrine brands because of client affordability concerns (Fig 5).

## Fidelity variation and moderating factors

Fidelity scores varied significantly by facility type (p = 0.04) and profession (p = 0.01). Medical doctors had the highest median scores. Scores increased significantly with more years of malaria case management experience (≥3 years) and overall service tenure (>10 years) (Table 4).

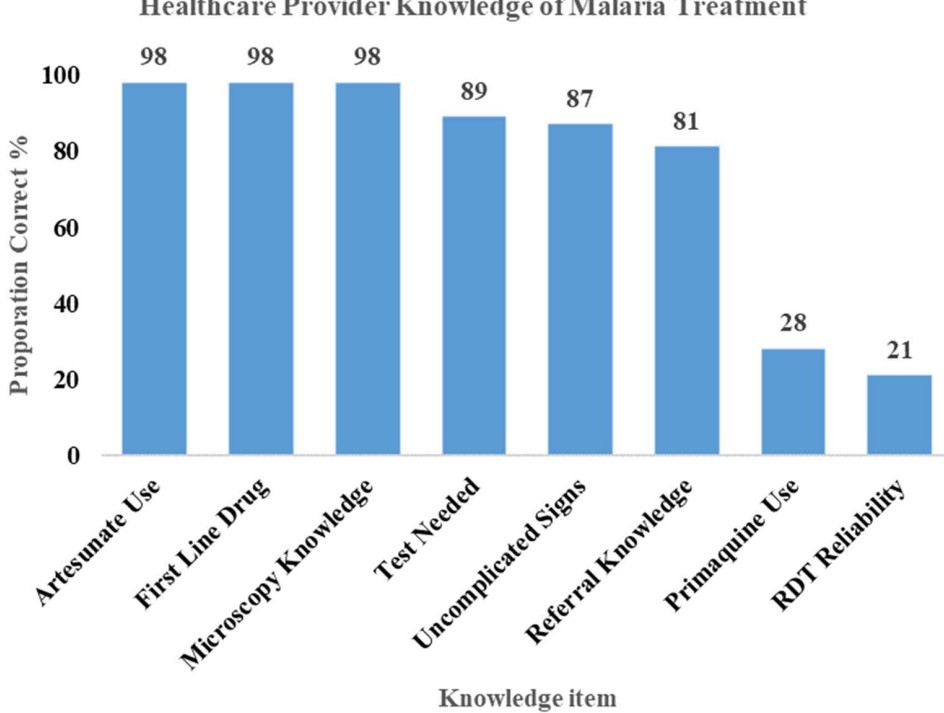

**Fig 3. Healthcare Providers' Knowledge of Malaria Treatments, Amhara Region, 2025.**

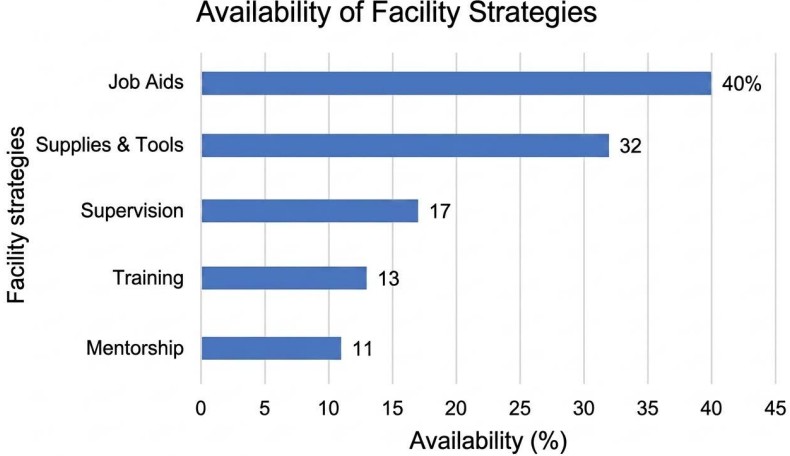

**Fig 4. Implementation of facility strategies, Amhara Region, 2025.**

## Factors associated with fidelity (multivariable models)

Multivariable linear regression identified several factors independently associated with implementation fidelity (Table 5). Higher fidelity scores were associated with greater participant responsiveness ($\beta = 3.4$, $p < 0.001$), stronger facilitation strategies ($\beta = 2.8$, $p < 0.001$), lower perceived intervention complexity (reverse-coded; $\beta = 2.1$, $p < 0.001$), better supply

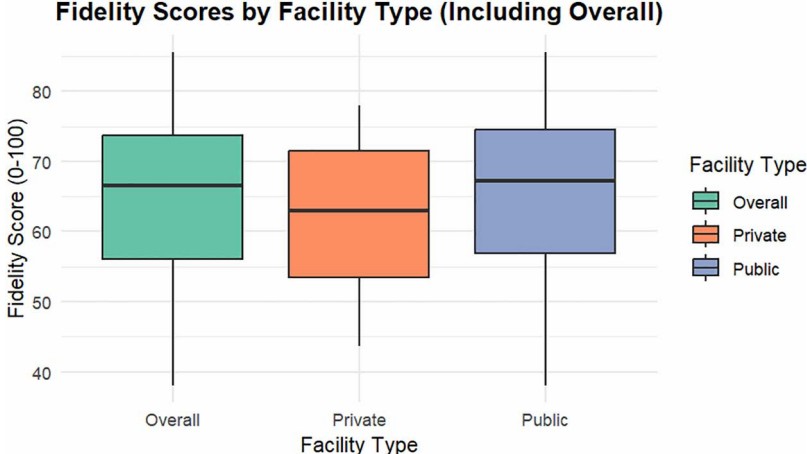

**Fig 5. Fidelity Score by Facility Type (n = 53), Amhara Region.2025. Box plots show median (line), IQR (box), and range (whiskers). Mann-Whitney U test: p = 0.041.**

**Table 4. Bivariate analysis of implementation fidelity scores.**

| Characteristic | n | Median Fidelity (IQR) | Test statistic | p-value |
|---|---|---|---|---|
| Sector | | | Z = −2.043 | 0.041* |
| Public | 38 | 67.0 (60.0–77.0) | | |
| Private | 15 | 63.0 (56.0–70.0) | | |
| Profession | | | $\chi^2$ = 9.215 | 0.010* |
| Medical Doctor | 9 | 74.0 (68.0–80.0) | | |
| Nurse | 24 | 62.0 (55.0–68.0) | | |
| Health Officer/Other | 20 | 65.0 (58.0–72.0) | | |
| Provider age group (years) | | | $\chi^2$ = 5.321 | 0.070 |
| <30 | 10 | 61.0 (55.0–68.0) | | |
| 30–39 | 36 | 65.0 (58.0–74.0) | | |
| 40–49 | 6 | 70.0 (63.0–76.0) | | |
| ≥50 | 1 | 72.0 (—) | | |
| National malaria guideline available on-site | | | Z = −2.31 | 0.021* |
| Yes | 36 | 68.0 (60.0–77.0) | | |
| No | 17 | 60.0 (54.0–67.0) | | |
| Diagnostic modality | | | $\chi^2$ = 4.87 | 0.088 |
| Microscopy only | 29 | 63.0 (56.0–70.0) | | |
| RDT only | 3 | 58.0 (52.0–65.0) | | |
| Both | 21 | 69.0 (62.0–78.0) | | |
| Stock out in preceding 3 months | | | Z = −1.89 | 0.059 |
| Yes | 22 | 61.0 (54.0–69.0) | | |
| No | 31 | 67.0 (60.0–76.0) | | |

*Statistically significant at p < 0.05. Mann-Whitney U test used for two-group comparisons; Kruskal-Wallis test used for three or more groups. IQR = interquartile range. The ≥50 age group contains a single observation; the median is reported without IQR

**Table 5. Multivariable linear regression (OLS): implementation moderators associated with composite fidelity score (0–100) (N = 53).**

| Predictor (moderator) | β (adjusted mean difference) | 95% CI | p-value |
|---|---|---|---|
| Participant responsiveness (scale) | 3.4 | (2.5, 4.3) | <0.001 |
| Facilitation strategies (scale) | 2.8 | (1.9, 3.7) | <0.001 |
| Intervention complexity (reverse-coded scale)* | 2.1 | (1.2, 3.0) | <0.001 |
| Supply access | 2.1 | (0.04, 4.1) | 0.045 |
| Supervision frequency | 1.8 | (0.5, 3.1) | 0.008 |

Intervention complexity was reverse-coded so that higher values indicate lower perceived complexity and easier implementation.

access ($\beta = 2.1$, $p = 0.045$), and more frequent supervision ($\beta = 1.8$, $p = 0.008$). Participant responsiveness showed the largest adjusted association with fidelity. The full model explained 58% of the variance in fidelity scores ($R^2 = 0.58$).

Facilitation strategies, including supervision, training, and resource availability, also significantly moderated fidelity scores ($\beta = 2.8$; $p < 0.001$). Among these, supportive supervision emerged as the most effective strategy, despite accounting for a relatively small portion (17%) of the overall implementation experience. Notably, human facilitation (e.g., mentoring, feedback) was less consistently available than material support (e.g., supplies, job aids), particularly in conflict-affected areas. Still, where supervision was maintained, fidelity outcomes improved markedly.

Intervention complexity was inversely related to fidelity. Simpler protocols were associated with significantly higher fidelity scores ($\beta = 2.1$; $p < 0.001$). Conversely, the higher levels of complexity, which are usually embraced whenever there is a resurgence because of the increase in case management needs and workload pressure, were associated with lower levels of adherence. This implies that the streamlining of guidelines can be of particular importance in times of high burden.

Supply Access ($\beta = 2.1$, $p = 0.045$) & Supervision Frequency ($\beta = 1.8$, $p = 0.008$) were also significant positive predictors, though challenged by conflict and supply chain issues.

A two-step hierarchical modeling approach was employed to identify factors associated with fidelity to national malaria guidelines during resurgence periods:

Model 1 (Contextual Factors Only): Facility and respondent-level covariates with a p-value < 0.20 in bivariate analysis were entered simultaneously and refined using backward elimination (exit criterion: $\alpha = 0.10$).

Model 2 (Full Model): Implementation-specific moderators, intervention complexity, participant responsiveness, and facilitation strategies were subsequently added to Model 1 to assess their adjusted associations with fidelity.

## Final model results

In the full model including facility and respondent characteristics, clinics and dispensaries had lower fidelity scores than hospitals, while health centers also showed a tendency toward lower fidelity, although this difference did not reach conventional statistical significance ($\beta = -1.2$, $p = 0.070$) (Table 6).

Fidelity scores were higher among providers aged 40–49 years than among those aged <30 years ($\beta = 2.1$, $p = 0.004$). Compared with medical doctors, nurses ($\beta = -2.8$, $p < 0.001$), laboratory personnel ($\beta = -3.0$, $p < 0.001$), and health officers ($\beta = -1.5$, $p = 0.040$) had lower fidelity scores. No evidence of problematic multicollinearity was observed, and the final model explained 58% of the variance in fidelity scores ($R^2 = 0.58$)

## Sensitivity analysis

To address potential social desirability bias in the self-reported frequency domain, we compared the original composite fidelity score with an adjusted score excluding the single self-reported frequency item. Excluding this item reduced the mean fidelity score from 64.3% to 61.2%, but the direction and significance of the public-private difference remained unchanged ($p = 0.038$ vs $p = 0.041$), supporting the robustness of the main findings (Fig 6).

**Table 6. Multivariable linear regression (OLS): facility characteristics, respondent characteristics, and moderators associated with composite fidelity score (0–100) (N = 53).**

| Variable | β (adjusted mean difference) | 95% CI | p-value |
|---|---|---|---|
| Moderators | | | |
| Intervention complexity (reverse-coded) | 2.1 | (1.2, 3.0) | <0.001 |
| Participant responsiveness | 3.4 | (2.5, 4.3) | <0.001 |
| Facilitation strategies | 2.8 | (1.9, 3.7) | <0.001 |
| Facility type | | | |
| Hospital | Ref | | |
| Health center | −1.2 | (−2.5, 0.1) | 0.070 |
| Clinic | −2.5 | (−4.0, −1.0) | 0.001 |
| Dispensary | −2.0 | (−3.3, −0.7) | 0.003 |
| Respondent age group | | | |
| <30 | Ref | | |
| 30–39 | 1.5 | (0.2, 2.8) | 0.020 |
| 40–49 | 2.1 | (0.7, 3.5) | 0.004 |
| ≥50 | 1.0 | (−0.5, 2.5) | 0.190 |
| Respondent profession | | | |
| Medical doctor | Ref | | |
| Health officer | −1.5 | (−3.0, −0.1) | 0.040 |
| Nurse | −2.8 | (−4.3, −1.3) | <0.001 |
| Laboratory technician/technologist | −3.0 | (−4.7, −1.3) | <0.001 |

β coefficients represent adjusted mean differences in the fidelity score (0–100) per unit increase in continuous predictors or relative to the reference category for categorical predictors. CI = confidence interval; Ref = reference category

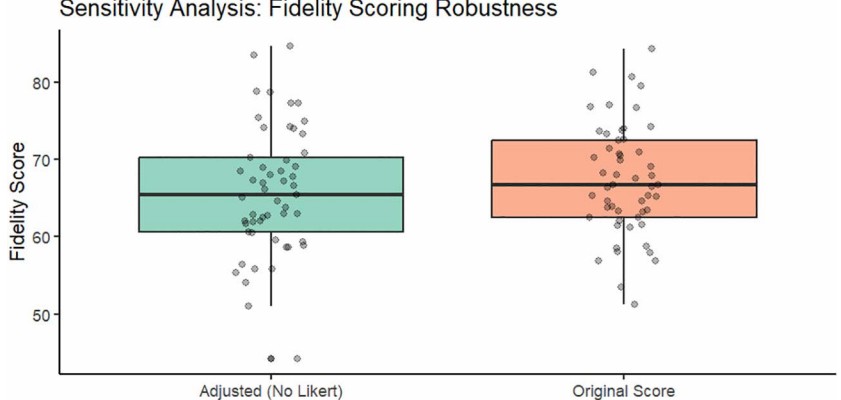

**Fig 6. Sensitivity analysis of fidelity scoring methods,** *comparison of mean fidelity using alternative scoring approaches.*

A joint display of quantitative fidelity findings and qualitative CFIR-informed barriers highlighted context-specific implementation challenges.

## Qualitative study findings

**Demographic characteristics.** In-depth interviews with 32 healthcare providers yielded two overarching themes, diagnostic fidelity and treatment fidelity, with nine related subthemes. These themes were subsequently mapped to CFIR

domains and constructs ([Table 1]). Among the qualitative interview participants (n = 32), mean years of service were higher in the private sector than in the public sector (8.2 vs 6.2 years) ([Table 7]). This pattern differed from the full quantitative sample, in which mean years of service were higher in the public sector (10.7 vs 7.5 years), reflecting the purposive sampling of experienced private-sector providers for in-depth interviews.

## Themes and Sub-themes

In-depth interviews with 32 healthcare providers from both the public and private sectors revealed two themes: adherence to diagnostic standards and adherence to treatment standards. Qualitative findings have identified two themes and nine sub-themes related to fidelity.

**Theme 1: Inner Setting Barriers to Diagnostic Fidelity (CFIR).**

a) **Microscopy**

Most healthcare providers stated that suspected malaria cases are screened for a blood test by microscopy before treatment, demonstrating strong adherence to the first-line diagnostic algorithm.

*"Almost all febrile case suspected patients are tested for malaria…The most common confirmed diagnosis is microscopy." (IDI; HF 4, Public facility, 38 Years)*

b) **Multispecies Rapid Diagnostic Tests (adaptive fidelity)**

A few healthcare providers reported preferring multispecies RDTs even when trained laboratory professionals were available, citing poor reagent quality and shortages of laboratory supplies. Although RDT use is allowed within the guideline, in these settings, it functioned as an adaptive response to supply-chain constraints.

**Table 7. Demographics of In-Depth Interview Participants by Facility Type (N = 32), Amhara Region, 2025.**

| Characteristic | Overall<br>N = 32[1] | Private<br>N = 10[1] | Public<br>N = 22[1] |
|---|---|---|---|
| Sex | | | |
| Female | 10 (31%) | 0 (0%) | 10 (45%) |
| Male | 22 (69%) | 10 (100%) | 12 (55%) |
| Age Group | | | |
| 20-29 | 10 (31%) | 1 (10%) | 9 (41%) |
| 30-39 | 12 (38%) | 9 (90%) | 3 (14%) |
| 40-49 | 9 (28%) | 0 (0%) | 9 (41%) |
| 50+ | 1 (3.1%) | 0 (0%) | 1 (4.5%) |
| Health Professional | | | |
| Laboratory technologist | 7 (22%) | 3 (30%) | 4 (18%) |
| Medical Doctor | 7 (22%) | 3 (30%) | 4 (18%) |
| Nurse | 11 (34%) | 4 (40%) | 7 (32%) |
| Public health officer | 7 (22%) | 0 (0%) | 7 (32%) |
| Years of Service | 6.8 (±2.5) | 8.2 (±2.8) | 6.2 (±2.1) |

[1] n (%); Mean (±SD)

Note: Values are n (%) unless otherwise indicated. This table reflects the purposively sampled qualitative subsample and is not intended to represent the survey sample.

*"…the quality of lab reagents is unreliable; I prefer multispecies RDT kits." (IDI-HF8, private facilities; 34years)*

### c) Presumptive/ Clinical Diagnosis (fidelity drift)

*Many healthcare providers adhered to test-before-treat protocols, but deviations were common. This reflects CFIR Inner Setting constraints in Available Resources (stockouts) and Structural Characteristics (lab quality), as well as Outer Setting pressures from Patient Needs & Resources (normative expectations for immediate treatment). As one provider stated:*

*"…Based on signs and symptoms, patients without diagnostic confirmation are sometimes treated as malaria." (IDI, HF 5, Public Facility, 38 years).*

*"For all suspected malaria patients, first perform a test. If the result is positive, proceed with treatment. However, in cases where symptoms are severe but the test result is negative, it may be necessary to treat as if the patient has malaria." (IDI-HF-12, Private facility, 39years)*

*"When frosted slides are out of stock, we resort to clinical diagnosis as patients often insist on receiving treatment." (IDI-HF20, public facility, 33 Years).*

*"… Experience can be very helpful, though never as a replacement for testing. Even senior doctors can make errors without confirmation." (IDI-HF 19, public facility, 38 Years).*

This behavior also aligns with Outer Setting patient expectations (perceived demand for immediate treatment), creating tension between guideline fidelity and client satisfaction

**Theme 2: Outer Setting Pressures on Treatment Fidelity (CFIR).**

### a) Correct Prescription & Dosing (partial fidelity)

Most healthcare providers reported adherence to national malaria treatment guidelines, including appropriate pre-scribing of regimens for Plasmodium falciparum and Plasmodium vivax. Despite the guideline recommendation to reserve chloroquine for P. vivax, a few healthcare providers indicated that they sometimes used Coartem to treat P. vivax cases.

*"An adult is treated with 10 chloroquine for P. vivax, but sometimes P. vivax cases are treated with Coartem." (IDI, HF 2, public facility, 37 years)*

b) This finding suggests partial fidelity: the correct class of antimalarial was often prescribed, but not always in a species-specific manner. Providers attributed this to species misclassification, diagnostic uncertainty, concerns about laboratory accuracy, and medicine availability. **Non-standard Drug Combinations (fidelity erosion)**

A few healthcare providers, most of whom were employed by privately held, profit-making organizations, mentioned using non-standard antimalarial drugs. This demonstrated non-adherence to national guidelines, as healthcare providers prior-itized business viability over adhering to a standardized treatment protocol since they were focused on client satisfaction regarding cost.

*"Even non-standard forms of artemether-lumefantrine are prescribed by me, since this is a private, for-profit business." (IDI, 35 years, Private Facility, HF 9)*

## c) Directly Observed Therapy (DOT) for First Dose

Most healthcare providers in both public and private facilities reported that they could not routinely ensure directly observed therapy for the first dose. This gap may compromise treatment quality, particularly when early vomiting occurs and repeat dosing is needed.

> *"Most of malaria positive cases take the first dose at home, which is not under the supervision at the facility…If they vomit, we can't readminister." (IDI, HF 10, Public Facility, 37 years)*

## d) Comprehensive Counseling & Follow-up (variable fidelity)

Most healthcare providers mentioned that the compliance with the counseling component of the national malaria treatment guideline was low in most facilities in both the private and the public sectors. Particularly, they tended to lack important details about the malaria transmission, the proper regimen, or red flags that needed to be treated immediately. Instead, patients were frequently referred to health extension workers (HEWs) for follow-up, resulting in fragmented communication and missed opportunities to reinforce critical health messages.

> *"We rarely provide counseling on key messages, and we seldom confirm if patients understood the information." (IDI, HF 29, Public Facility, 42 years)*

## e) Follow-Up Mechanisms

Many healthcare providers reported that no formal mechanism existed to follow patients after treatment initiation, especially once they had left the facility.

> *"We treat them and advise them to come back in case it becomes worse, but we never follow up or make sure that they took the dose. (IDI, HF 27, public facility, 37 years)*

> *"We do not know at all whether they complete the treatment or whether they are taking it. That's left to the patients themselves." (IDI, HF 21, private facility, 35 years)*

## f) Patient Involvement

Many healthcare providers reported not routinely discussing diagnoses or treatment plans with patients. This limited communication contributes to poor patient compliance, especially when diagnostic results are negative, but symptoms persist. Such gaps in provider-patient interaction and test adherence may undermine test-and-treat strategies, potentially fuel inappropriate treatment, and increase the risk of malaria resurgence.

> "I didn't explain the diagnosis and treatment plan… patients still prefer injections or traditional remedies." *(IDI, HF 05, public facility, 35 years)*

> "Sometimes patients insist on treatment even if the test is negative. Educating them is part of our role, but it's not always easy." *(IDI, HF 03, private facility, 35 years)*

**Fidelity gaps and determinants (CFIR-informed)**

Theme 1: Diagnostic fidelity constrained by resources and workflow (Inner setting; outer setting pressure)

Providers described strong intention to follow test-before-treat but noted deviations during reagent shortages, high workload, or when patients demanded immediate treatment. Some reported using multispecies RDTs as an adaptive response to inconsistent microscopy supplies or perceived quality issues.

Theme 2: Treatment fidelity undermined by non–species-specific prescribing, limited observed dosing, and weak counseling/follow-up (Characteristics of individuals; inner setting; outer setting)

Providers commonly reported guideline-concordant first-line treatment knowledge but described challenges implementing species-specific regimens when key medicines (e.g., primaquine) were unavailable or when diagnostic certainty was questioned. Many facilities did not routinely observe the first dose, and counseling/follow-up mechanisms were inconsistent.

### Integration of quantitative and qualitative findings

Joint displays indicated convergence between quantitative evidence of moderate fidelity and qualitative explanations emphasizing supply constraints, diagnostic uncertainty, patient pressure, limited supervision/mentorship, and challenges implementing complex steps during surge conditions. Divergence was noted between relatively high self-reported testing frequency and qualitative accounts suggesting inconsistent follow-up and counseling, supporting the decision to include sensitivity analyses excluding self-reported frequency (Table 8 and Fig 7.)

## Discussion

This mixed-methods study assessed fidelity to malaria test-and-treat guidelines in Ethiopia's Amhara Region during a significant resurgence. Applying Carroll's fidelity framework and CFIR, we identified moderate overall fidelity (64.3%), with public facilities outperforming private ones. Key drivers included provider responsiveness, facilitation strategies (especially supervision), and lower intervention complexity.

**Table 8. Joint display of fidelity domains/moderators: quantitative results and qualitative explanations.**

| Construct/domain | Quantitative finding | Qualitative explanation (themes) | Integrated interpretation |
|---|---|---|---|
| Content (diagnostic + treatment steps) | Only 51% showed evidence consistent with species-specific prescribing; non-standard AL brands reported in some private facilities | Diagnostic uncertainty, medicine availability (e.g., primaquine), patient affordability, and private-sector incentives contributed to non–species-specific or non-standard treatment | Convergent evidence of content/adherence gaps, more salient in private settings |
| Coverage (tested before treatment) | Composite fidelity moderate overall; sector gap (public > private) | Stockouts of lab consumables and workflow delays led to occasional presumptive treatment; patient pressure influenced deviations | Coverage gaps largely linked to resource constraints and external pressure |
| Frequency (consistent testing) | Self-reported testing frequency contributed to the composite score | Interviews emphasized inconsistent observed first dose, counseling, and lack of follow-up; some providers described intermittent testing during shortages | Apparent divergence likely reflects self-report bias and differences between "testing frequency" vs "continuity of care" |
| Participant responsiveness | Strong positive association with fidelity ($\beta = 3.4$; $p < 0.001$) | Providers described motivation/ownership enabling improvisation and adherence under constraints | Strong convergence: engagement/ownership supports fidelity maintenance |
| Facilitation strategies | Positive association with fidelity ($\beta = 2.8$; $p < 0.001$); supervision/training less common than job aids | Limited mentorship/supervision, particularly in private facilities and disrupted areas | Convergent: strengthening supervision/mentorship likely improves fidelity |
| Intervention complexity | Lower perceived complexity associated with higher fidelity ($\beta = 2.1$; $p < 0.001$) | Providers described guideline steps as difficult during surges/high workload | Convergent: simplified job aids/surge algorithms may protect fidelity |

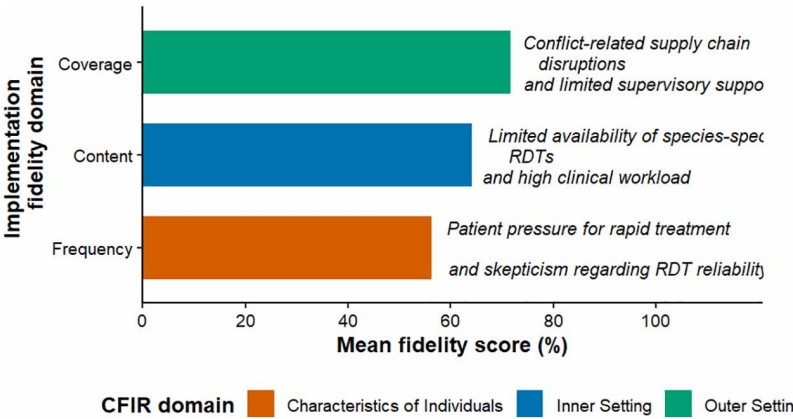

**Fig 7. Joint display of quantitative fidelity scores and qualitative CFIR barriers, matrix visualizing facility-level fidelity scores alongside identified implementation barriers.**

The study revealed that public health facilities demonstrated higher compliance compared to private ones. These findings underscore the urgent need to enhance malaria control efforts and bolster health system resilience in conflict-affected and resource-limited settings.

This study confirms that public facilities demonstrated superior diagnostic capability, utilizing microscopy and RDTs more comprehensively than private facilities reliant on single methods. This aligns with national guidelines prioritizing microscopy for species identification and a previous study [45]. The finding that presumptive treatment persisted, particularly in private settings and during stockouts, mirrors challenges documented in other low-resource areas [46] and highlights a critical barrier to test-before-treat fidelity. Adaptive practices like using multispecies RDTs due to reagent shortages, while pragmatic, represent deviations from protocol [25,47].

This study indicated that despite high knowledge, species-specific prescribing remained suboptimal; interviews indicated supply constraints (e.g., primaquine availability) and diagnostic uncertainty. Exhaustion of primaquine (needed to treat P. vivax) meant providers used AL on all the species even though they knew it was inappropriate, a condition known as compromise fidelity being exhibited in conflict areas

This study revealed that Public-sector healthcare providers had longer work experience, better tools, and more training opportunities compared to private-sector healthcare providers, consistent with the previous study from Nigeria [25]. However, guidelines were more accessible in public facilities, unlike in some areas, like Gambela, where local healthcare providers lack access to these documents [20]. The study indicated that Healthcare providers cited shortages of reagents, time constraints, and patient demands as reasons for treating patients without proper testing, similar to findings in other low-resource areas [46]. The study highlighted cases of adaptive fidelity, including the use of multispecies RDTs prompted by reagent shortages, reflecting issues raised in implementation studies from Nigeria and Kenya [25,48].

This study indicated treatment fidelity revealed significant gaps. While first-line drugs were generally prescribed, species-specific adherence was poor (only 51%), with *P. vivax* often incorrectly treated with artemether-lumefantrine (AL). This finding, consistent with other Ethiopian studies [22], risks inadequate radical cure for *vivax* and may contribute to relapse transmission. The use of non-standard AL brands in private facilities (11%), driven by affordability concerns [20], a business-model incentive skewed with public health goals [49]. Further compromises treatment quality and potentially fuels drug resistance. Crucially, Directly Observed Therapy (DOT) for the first dose was rarely implemented, and follow-up mechanisms were largely absent. These gaps, undermining treatment efficacy and surveillance [19,47], represent major weaknesses in the implementation cascade.

This study revealed Facilitation strategies (especially supportive supervision) significantly enhanced fidelity, but implementation was uneven, mirroring studies in other parts of Ethiopia [50] and Nigeria [51]. The study also mentioned that job aids and supplies were common, but intensive supports (like mentorship and regular training) were infrequent, especially in conflict zones, consistent with previous studies in Uganda [52] and reviews the report in sub-Saharan Africa [53] and in Kenya [54].

This study mentioned that fidelity scores were significantly higher in public than private facilities, consistent with earlier studies in Ethiopia and elsewhere that show public sector alignment with donor-supported programs and vertical supervision structures [55–57].Conversely, private healthcare providers reported limited access to national guidelines, fewer diagnostic tools, and greater autonomy in decision-making. This autonomy, while allowing flexibility, may also facilitate guideline drift in pursuit of client satisfaction or profitability, thereby undermining standardization efforts. The finding that medical doctors achieved the highest fidelity scores aligns with evidence linking specialized training and decision-making autonomy to better guideline adherence [54,58], while lower scores among nurses and lab personnel may reflect training gaps or insufficient confidence with complex protocols [59].

This study revealed that the resurgence context, marked by conflict, supply chain disruptions, and facility damage [12,60], directly impeded fidelity through stockouts (reported by 41% of facilities) and limited supervision/training access. Despite this, facilities with engaged healthcare providers and consistent supervision maintained better fidelity, underscoring the resilience fostered by strong human factors and support systems, as advocated by WHO [8].

This study described in multivariable analysis confirmed participant responsiveness (provider engagement) as the strongest fidelity predictor (β=3.4, p<0.001), consistent with implementation science [61–63]. Higher responsiveness was associated with higher fidelity; qualitative findings suggested proactive problem-solving. Facilitation strategies, particularly supportive supervision (though only available in 17% of facilities), were also crucial (β=2.8, p<0.001), echoing studies emphasizing its value in Ethiopia [50] and Africa [51,53]. The inverse relationship between perceived intervention complexity and fidelity (β=2.1, p<0.001) highlights the vulnerability of complex protocols during high-burden periods or in resource-poor settings [28,33] Streamlining guidelines during surges may be essential. Historical examples of malaria resurgence illustrate that overly intricate or rapidly evolving guidelines, combined with limited preparation and training, can generate confusion, selective adoption, or outright rejection [64].

Although the self-reported Frequency domain was vulnerable to recall and social desirability bias, the sensitivity analysis showed that excluding this domain did not change the direction or significance of the public-private difference.

This study illustrates the value of mixed-methods implementation research. Quantitative findings identified moderate fidelity and key correlates, while qualitative findings clarified mechanisms, such as stockouts, diagnostic distrust, patient pressure, and communication failures, that plausibly drive observed fidelity gaps [46,51,65]. The near-universal failure to implement DOT and follow-up, alongside poor patient counseling, reveals systemic weaknesses in care continuity and community engagement [19,22,47], threatening treatment outcomes and resurgence control [66]. The divergence between self-reported frequency measures and interview accounts of inconsistent follow-up and DOT highlights the importance of triangulating self-reported implementation indicators with qualitative evidence and record-based measures [28,67].

## Strengths and limitations

Strengths include the use of established implementation frameworks, concurrent quantitative and qualitative data collection, and comparison across public and private sectors. Limitations include reliance on a self-reported frequency measure, which is susceptible to social desirability bias; however, sensitivity analyses excluding frequency did not alter the public-private difference. The facility sample size limited model complexity; we therefore used parsimonious, prespecified models. Findings reflect Amhara during a resurgence context and may not generalize to non-resurgence settings.

## Conclusion

Implementation fidelity to Ethiopia's malaria test-and-treat guideline in the Amhara Region during the 2024–2025 resurgence was moderate, with lower fidelity in private facilities than in public facilities. Fidelity gaps were characterized by

departures from test-before-treat practices under supply and workload constraints, incomplete species-specific prescribing, limited supportive supervision, and occasional use of non-standard antimalarial brands in the private sector. Provider responsiveness, facilitation strategies (particularly supportive supervision), and lower perceived intervention complexity were independently associated with higher fidelity.

To strengthen guideline implementation during resurgences, malaria programs should prioritize: (i) supportive supervision and mentorship with explicit inclusion of private facilities; (ii) targeted, scenario-based refresher training focused on species-specific management, RDT interpretation, and patient communication; (iii) strengthened supply-chain reliability for diagnostics and quality-assured medicines; and (iv) simplified job aids and surge-adapted decision supports that are consistent with national guidance. Routine monitoring of fidelity indicators—along with implementation determinants such as provider responsiveness, may help identify early implementation failures and support rapid corrective action. These findings support Ethiopia's 2023–2030 malaria strategy priorities on private-sector engagement and outbreak resilience by identifying context-specific barriers to guideline implementation during resurgence.

## Supporting information

**S1 Fig. Sampling flow diagram for selection of woredas and health facilities.** Flow diagram showing the multi-stage sampling procedure used to select 53 health facilities (38 public and 15 private) from 19 woredas in Amhara Region, Ethiopia, in 2025. Of 236 woredas in the region, 166 had complete weekly malaria surveillance data and were screened for resurgence; 30 met the resurgence criterion, defined as a ≥ 50% increase in confirmed malaria cases in 2024 compared with 2022. Selected woredas were stratified by geographic zone, and facilities were sampled separately for the public and private sectors.
(DOCX)

**S1 Table. Implementation fidelity measurement framework and operationalization.** Item-level specification of the composite fidelity score, including domain definitions, data sources, recall windows, scoring rules, and standardization procedures.
(DOCX)

**S1 File. Structured questionnaire for implementation fidelity assessment.** KoboToolbox questionnaire adapted from Carroll's implementation fidelity framework and WHO malaria guidelines, covering provider and facility characteristics, fidelity domains (content, coverage, and frequency), and CFIR-aligned implementation moderators. The questionnaire was pilot tested in 3 facilities before data collection.
(DOCX)

**S2 Table. Semi-structured interview guide for healthcare providers.** Topic guide used for in-depth interviews, organized by CFIR-aligned domains. The guide was pretested with 2 providers, administered in Amharic, and translated into English as described in the Methods.
(DOCX)

## Acknowledgments

We would like to thank the study participants and health facilities for allowing the data to be collected. We also thank Mr. Melaku for technical support in developing the KoboToolbox survey and data verification tools.

## Author contributions

**Conceptualization:** Mastewal Worku Lake.

**Data curation:** Mastewal Worku Lake.

**Formal analysis:** Mastewal Worku Lake.

**Methodology:** Mastewal Worku Lake, Muluken Azage Yenesew.

**Supervision:** Mastewal Worku Lake, Muluken Azage Yenesew.

**Validation:** Mastewal Worku Lake, Kassahun Alemu Gelaye, Mulusew Andualem Asemahagn, Kindie Fentahun Muchie, Hailemariam Awoke Engedaw, Muluken Azage Yenesew.

**Writing – original draft:** Mastewal Worku Lake.

**Writing – review & editing:** Mastewal Worku Lake, Kassahun Alemu Gelaye, Mulusew Andualem Asemahagn, Kindie Fentahun Muchie, Hailemariam Awoke Engedaw, Muluken Azage Yenesew.

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
