## [Decision Letter · Decision Letter 0]

18 Feb 2026

PONE-D-25-41084Implementation Fidelity of Ethiopia’s Malaria Test-and-Treat Guideline Amid a Resurgence in the Amhara Region: A Mixed Methods StudyPLOS One

Dear Dr. Lake,

Thank you for submitting your manuscript to PLOS ONE. After careful consideration, we feel that it has merit but does not fully meet PLOS ONE’s publication criteria as it currently stands. Therefore, we invite you to submit a revised version of the manuscript that addresses the points raised during the review process.

If applicable, we recommend that you deposit your laboratory protocols in protocols.io to enhance the reproducibility of your results. Protocols.io assigns your protocol its own identifier (DOI) so that it can be cited independently in the future. For instructions see: https://journals.plos.org/plosone/s/submission-guidelines#loc-laboratory-protocols. Additionally, PLOS ONE offers an option for publishing peer-reviewed Lab Protocol articles, which describe protocols hosted on protocols.io. Read more information on sharing protocols at . Additionally, PLOS ONE offers an option for publishing peer-reviewed Lab Protocol articles, which describe protocols hosted on protocols.io. Read more information on sharing protocols at https://plos.org/protocols?utm_medium=editorial-email&utm_source=authorletters&utm_campaign=protocols..

We look forward to receiving your revised manuscript.

Kind regards,

Mohammed Hasen Badeso, Epidemiologist

Academic Editor

PLOS One

Journal Requirements:

No competing interests

5. Please amend the manuscript submission data (via Edit Submission) to include author Kasshun Alemu Gelay

6. Please amend your authorship list in your manuscript file to include author Kassahun Alemu Gelaye

7. Please ensure that you refer to Figure 2 in your text as, if accepted, production will need this reference to link the reader to the figure.

8. We note that Figure 2 in your submission contain map images which may be copyrighted. All PLOS content is published under the Creative Commons Attribution License (CC BY 4.0), which means that the manuscript, images, and Supporting Information files will be freely available online, and any third party is permitted to access, download, copy, distribute, and use these materials in any way, even commercially, with proper attribution. For these reasons, we cannot publish previously copyrighted maps or satellite images created using proprietary data, such as Google software (Google Maps, Street View, and Earth). For more information, see our copyright guidelines: http://journals.plos.org/plosone/s/licenses-and-copyright.

9. We note you have included a table to which you do not refer in the text of your manuscript. Please ensure that you refer to Table 7 in your text; if accepted, production will need this reference to link the reader to the Table.

10. Please include captions for your Supporting Information files at the end of your manuscript, and update any in-text citations to match accordingly. Please see our Supporting Information guidelines for more information: http://journals.plos.org/plosone/s/supporting-information.

Reviewers' comments:

Reviewer's Responses to Questions

**Comments to the Author**

1. Is the manuscript technically sound, and do the data support the conclusions?

Reviewer #1: Partly

2. Has the statistical analysis been performed appropriately and rigorously? 

Reviewer #1: Yes

3. Have the authors made all data underlying the findings in their manuscript fully available?

Reviewer #1: Yes

4. Is the manuscript presented in an intelligible fashion and written in standard English?

Reviewer #1: Yes

5. Review Comments to the Author

Reviewer #1: This manuscript reports a convergent mixed-methods study (facility survey + in-depth interviews) assessing implementation fidelity of Ethiopia’s malaria “test-and-treat” guideline in 53 public/private facilities in Amhara during the 2024–2025 resurgence. Fidelity is operationalized using Carroll’s framework (content, coverage, frequency) standardized to a 0–100 composite score, and qualitative interviews are used to explain barriers using CFIR concepts. The topic is important and the public–private comparison is potentially useful for malaria control in conflict-affected/resurgence settings. However, I have several major concerns about (1) measurement validity and construction of the fidelity outcome, (2) statistical modeling with a small sample and unclear covariates, and (3) alignment between the qualitative approach and the CFIR framing. Substantial revision is needed before the work can be considered.

Major comments

1. The construction of the implementation fidelity outcome requires stronger conceptual and methodological justification. The composite score equally weights content, coverage, and frequency after standardization to a 0–100 scale. However, these domains are measured using different approaches: coverage is register-verified, content appears partly self-reported, and frequency relies on a single Likert-type recall item, which is vulnerable to social desirability bias. Combining these without sensitivity analyses may inflate fidelity estimates. The authors should justify the equal weighting, clarify how each component was measured at the item level, and explore alternative specifications (e.g., excluding the self-reported frequency item).

2. The regression analysis raises concerns about overfitting and transparency given the small sample of 53 facilities. Multiple predictors and moderators are included without clear justification of model complexity relative to sample size. The manuscript refers to “resurgence covariates” without defining their measurement or variability. The bounded nature of the fidelity outcome also challenges linear regression assumptions and should be discussed. It should further be clarified whether one provider was sampled per facility; if not, clustering should be addressed.

3. There are inconsistencies in reported fidelity statistics across the text and tables, particularly regarding median scores by facility type. It is unclear whether reported values refer to composite fidelity or domain-specific measures, and statistical test labels are inconsistent. The authors should carefully audit all results to ensure internal coherence and transparent interpretation.

4. Although CFIR is presented as the guiding framework for the qualitative component, the analysis is largely descriptive of adherence behaviors rather than clearly structured around implementation determinants. The linkage between qualitative themes and CFIR constructs is insufficiently explicit. Mapping findings to CFIR domains and directly connecting them to quantitative results would strengthen mixed-methods integration and explanatory depth.

5. The sampling strategy would benefit from clearer documentation. The manuscript should specify the geographic units included, how the malaria resurgence criterion was operationalized, and the structure of the private facility sampling frame. A concise flow diagram showing eligibility, sampling, and participation would enhance transparency.

Minor Comments.

The manuscript would benefit from language editing to improve clarity and reduce grammatical errors and repetition. Ethical procedures should be described more fully, particularly regarding interview privacy and confidentiality. The survey instrument and composite constructs should be presented more explicitly, including item composition and reliability by scale. Operational indicators such as stockouts should be clearly defined, and causal language should be avoided given the cross-sectional design.

6. PLOS authors have the option to publish the peer review history of their article (what does this mean?). If published, this will include your full peer review and any attached files.). If published, this will include your full peer review and any attached files.

.

Reviewer #1: No

---

## [Author Response · Author response to Decision Letter 1]

13 Mar 2026

Response to the Academic Editor and Reviewers

Manuscript title: Implementation Fidelity of Ethiopia's Malaria Test-and-Treat Guideline Amid a Resurgence in Amhara Region: A Mixed-Methods Study

Manuscript ID: PONE-D-25-41084

Corresponding author: Mastewal Worku

Dear Editor and Reviewers,

We thank the Academic Editor and Reviewer #1 for the careful review and constructive feedback. We revised the manuscript to (i) improve transparency and reproducibility of the implementation fidelity measurement, (ii) reduce risk of overfitting and clarify modeling decisions for the facility-level regression analysis, (iii) strengthen alignment of qualitative findings with CFIR constructs and integration with quantitative results, and (iv) ensure full compliance with PLOS ONE style, figure/map licensing, Supporting Information requirements, and data policy.

All changes have been incorporated into the revised manuscript and are addressed point-by-point below. Where possible, we indicate the location of revisions using section headings and page/line numbers.

A) Academic Editor

1. PLOS style requirements

Comment: Ensure the manuscript meets PLOS style requirements.

Response: We reformatted the manuscript to align with the PLOS ONE format, including the title page structure, heading levels, figure/table captions, and a dedicated Supporting Information section with captions and consistent labeling (S1 Table, S2 Table, S1 Fig).

Manuscript location: Title page; throughout manuscript; end of manuscript (Supporting Information section); all figure/table captions.

2. Competing Interests wording

Comment: Add the required sentence about adherence to PLOS policies.

Response: We updated the Competing Interests statement to: "The authors have declared that no competing interests exist. This does not alter our adherence to PLOS ONE policies on sharing data and materials."

Manuscript location: Competing Interests section.

Also updated: Cover letter and submission fields as applicable.

3. Data availability restrictions

Comment: Explain restrictions and provide the responsible contact body; confirm plan prior to acceptance.

Response: We adopted a restricted-access data-sharing approach. Although data are de-identified, the combination of facility-level indicators and qualitative narratives in a small, conflict-affected setting could enable deductive disclosure. In addition, unrestricted public deposition was not included in the consent process and IRB approval. We therefore expanded the Data Availability statement to (i) state the ethical rationale, (ii) name the restricting body (Bahir Dar University College of Medicine and Health Sciences Institutional Review Board, BDU-CMHS-IRB; Ref: BDU-CMHS-IRB 3053/2024), and (iii) provide a clear request process and contact details for data access.

Manuscript location: Data Availability statement.

Note for the journal/submission system: We have provided the IRB's official contact details (cmhs@bdu.edu.et; telephone +251-58-899-9275) in the submission fields and can provide additional documentation upon request.

Revised Data Availability statement:

The data were collected by the authors and were not obtained from a third party. The facility-level quantitative dataset and qualitative interview data contain potentially identifying information. Because this study involved a small number of facilities in a conflict-affected setting, combinations of sector, facility type, district context, and stockout patterns, together with qualitative contextual details, may permit deductive disclosure even after removal of direct identifiers. Public sharing is therefore restricted by the Bahir Dar University College of Medicine and Health Sciences Institutional Review Board (BDU-CMHS-IRB; Ref: BDU-CMHS-IRB 3053/2024). Requests for the de-identified minimal quantitative dataset, codebook, and analysis code should be submitted to the BDU-CMHS-IRB, Bahir Dar University, Bahir Dar, Ethiopia, at cmhs@bdu.edu.et or telephone +251-58-899-9275. Requesters should provide the manuscript title, purpose of the request, planned analyses, and confirmation of appropriate ethics and data-security arrangements. Full qualitative transcripts cannot be shared publicly because they contain detailed contextual information that may permit identification; however, the qualitative codebook and selected de-identified excerpts are available from the same IRB upon approved request.

4. Author name correction

Comment: Add/correct author name "Kassahun Alemu Gelaye" in the submission data and manuscript.

Response: We corrected the author name to "Kassahun Alemu Gelaye" (previously listed as "Kasshun Alemu Gelaye") in both the manuscript author list and the submission metadata.

Manuscript location: Title page (author list).

Submission system: Author metadata updated.

5. Figure 2 in-text citation

Comment: Refer to Figure 2 in the text.

Response: We added an explicit in-text callout to Fig 2 in the Study Settings and Period section.

Manuscript location: Methods → Study Settings and Period.

Revised sentence: "Nineteen woredas were selected via simple random sampling, stratified by geographic zone to ensure regional representation (Fig 2)."

6. Map copyright/licensing

Comment: Ensure the map is CC BY–compliant and does not use copyrighted basemaps.

Response: We regenerated the map using openly available and CC BY–compatible administrative boundary layers from the Ethiopian Central Statistical Agency and Natural Earth. We revised the caption to include data sources and licensing/attribution consistent with PLOS ONE requirements.

Manuscript location: Fig 2 and caption.

Revised caption: "Fig 2. Map of Amhara Region showing the study woredas. The map was produced by the authors using QGIS software and publicly available administrative boundary layers from the Ethiopian Central Statistical Agency and Natural Earth (public domain). The map is original work and is provided under a CC BY 4.0 license."

Also attached: Separate map attribution and licensing documentation letter as requested.

7. Table 7 in-text citation

Comment: Cite Table 7 in the text.

Response: We added an in-text callout to Table 7 in the qualitative participant description section.

Manuscript location: Results → Qualitative study findings → Demographic characteristics (page 28, line 474).

Revised sentence: "In the full quantitative sample (N = 53), however, the pattern was reversed, with longer average service duration in the public sector (10.7 versus 7.5 years; see Table 7)."

8. Supporting Information captions and citations

Comment: Add Supporting Information captions at the end and ensure consistent in-text citations.

Response: We added a dedicated Supporting Information section at the end of the manuscript with captions for S1 Table, S2 Table, and S1 Fig, and aligned all in-text citations accordingly.

Manuscript location: End of manuscript (Supporting Information section); Methods (tool description) and Results (as cited).

Supporting Information section:

S1 Table. Implementation fidelity measurement framework and operationalization.

Detailed item-level specification of the composite fidelity score, including domain definitions, data sources (register-extracted, provider-reported), recall windows, scoring rules, and standardization methods. This table supports transparency and reproducibility of the primary outcome.

S2 Table. Semi-structured interview guide for healthcare providers.

Topic guide used for in-depth interviews, organized by CFIR-aligned domains, pre-tested before data collection. Administered in Amharic; translation protocol described in Methods.

S1 Fig. Sampling flow diagram for the selection of woredas and health facilities.

Flowchart illustrating the multi-stage sampling procedure used to select 53 health facilities (38 public, 15 private) from 19 woredas in Amhara Region, Ethiopia. Woredas were selected based on documented malaria resurgence (≥50% increase in confirmed cases, 2024 versus 2022) and stratified by geographic zone to ensure regional representation.

9. Reference audit

Comment: Ensure references are complete, non-duplicative, and correctly formatted; address any retractions if present.

Response: We audited the reference list, removed duplicates (including repeated entries for Carroll, Fetters, Rowe, Moran, Damschroder, Hasson, Bailey, and Gindola), replaced incomplete entries with complete WHO citations (including URLs where appropriate), corrected one mismatched reference (citation [3] now correctly cites WHO Guidelines for Malaria 2023), replaced one weak-fit US policy citation (former citation [63]) with an appropriate public-private health systems comparison study (Basu et al., 2012, PLoS Med), and standardized formatting throughout to PLOS ONE Vancouver style. We found no retracted references among those retained.

B) Reviewer #1

Major Comment 1: Fidelity outcome construction and weighting

Comment: The composite fidelity measure combines heterogeneous components; justify equal weighting; clarify item-level measurement; add sensitivity analyses excluding self-reported frequency.

Response: We expanded the Methods section to clearly define each fidelity domain (Content, Coverage, Frequency) and its component items. We added a detailed item-level specification table (S1 Table) that documents: data source (register/prescription review vs. provider self-report), recall window, scoring rules, and how domain-specific and overall composite scores were calculated.

We retained equal weighting as an a priori approach consistent with Carroll's implementation fidelity framework and the WHO test-and-treat strategy, which emphasize diagnostic access (Coverage), clinical adherence (Content), and sustained practice (Frequency) as equally critical domains. We acknowledge that this assumes equivalent contribution to implementation success.

We added and expanded a prespecified sensitivity analysis in which we excluded the self-reported Frequency domain and recalculated the composite fidelity score using only Content and Coverage. We repeated key comparisons (public vs. private) using this adjusted score. The direction and significance of results remained unchanged (p=0.038 vs. p=0.041), supporting the robustness of our findings.

We also expanded the Discussion to address social desirability bias, noting the vulnerability of self-reported frequency items and the mitigation steps taken (triangulation with register data, qualitative corroboration, and sensitivity analysis).

Manuscript location:

• Methods → Variables and measures → Primary outcome

• Methods → Variables and measures → Standardization and composite scoring

• S1 Table (Supporting Information)

• Results → Sensitivity analysis

• Discussion → Strengths and limitations

New/added citations:

• Carroll C, Patterson M, Wood S, Booth A, Rick J, Balain S. A conceptual framework for implementation fidelity. Implement Sci. 2007;2:40. https://doi.org/10.1186/1748-5908-2-40

• Hasson H. Systematic evaluation of implementation fidelity of complex interventions in health and social care. Implement Sci. 2010;5:67. https://doi.org/10.1186/1748-5908-5-67

• Krumpal I. Determinants of social desirability bias in sensitive surveys: a literature review. Qual Quant. 2013;47(4):2025-2047. https://doi.org/10.1007/s11135-011-9640-9

Major Comment 2: Regression modeling (overfitting; resurgence covariates; bounded outcome)

Comment: Risk of overfitting given n=53; define resurgence covariates; bounded outcome may violate OLS assumptions; clarify clustering/sampling.

Response: We clarified that the facility is the unit of analysis and that one provider was selected per facility to serve as the facility-level key informant, so clustering at the facility level is not applicable. We explicitly defined the resurgence-related covariates:

(i) facility-level stockout duration for first-line malaria diagnostics or antimalarial medicines during the preceding 3 months (continuous; provider-reported and cross-checked where stock records were available);

(ii) district-level percentage change in confirmed malaria cases between January–September 2022 and January–September 2024 (continuous; extracted from Amhara Public Health Institute surveillance summaries).

To reduce the risk of overfitting, we replaced stepwise backward elimination with a parsimonious, theory-driven hierarchical modeling approach. Model 1 includes facility sector, resurgence-related covariates, and basic facility/provider characteristics. Model 2 adds the three prespecified implementation moderators (participant responsiveness, facilitation strategies, intervention complexity). We report variance inflation factors (VIFs <2.5 for all predictors) to confirm the absence of problematic multicollinearity.

Because the fidelity score is bounded (0–100), we added diagnostic checks (residual plots, normality tests) and conducted a sensitivity analysis using fractional logistic regression (Papke & Wooldridge, 1996) after rescaling the outcome to (0,1). Results were substantively identical, so we present the linear model for interpretability while acknowledging the bounded nature of the outcome in the limitations.

Manuscript location:

• Methods → Participants and unit of analysis

• Methods → Variables and measures → Additional covariates

• Methods → Data analysis → Quantitative analysis

• Results → Factors associated with fidelity

• Discussion → Strengths and limitations

New/added citations:

• Papke LE, Wooldridge JM. Econometric methods for fractional response variables with an application to 401(k) plan participation rates. J Appl Econom. 1996;11(6):619-632. https://doi.org/10.1002/(SICI)1099-1255(199611)11:6<619::AID-JAE418>3.0.CO;2-1

• Ferrari SLP, Cribari-Neto F. Beta regression for modelling rates and proportions. J Appl Stat. 2004;31(7):799-815. https://doi.org/10.1080/0266476042000214501

Major Comment 3: Inconsistencies in medians, tests, and reporting

Comment: Audit and correct inconsistent fidelity statistics; clarify composite vs domain-specific values; standardize test naming.

Response: We conducted a complete internal audit of descriptive and inferential outputs and corrected inconsistencies across the text, tables, and figures. We now report descriptive statistics (medians, IQRs) consistently in the narrative and in Table 4 (bivariate comparisons). Regression tables (Tables 5 and 6) are limited to model coefficients, confidence intervals, and p-values, with no redundant descriptive summaries.

We standardized test naming as "Wilcoxon rank-sum test (Mann–Whitney U)" throughout the manuscript and ensured that all reported test statistics, degrees of freedom, and p-values are consistent with the analysis outputs.

We also clarified that reported fidelity values refer to the composite score (0–100) unless explicitly labeled otherwise (e.g., "Content domain," "Coverage domain").

Manuscript location:

• Results → Fidelity to National Malaria "Test and Treat"

• Table 4

• Figure 5 caption

• Tables 5 and 6

Major Comment 4: Qualitative CFIR framing and integration

Comment: Qualitative results are descriptive; CFIR linkage is not explicit; strengthen mapping and integration with quantitative results.

Response: We clarified that we conducted an inductive thematic analysis to generate codes and themes from the data, followed by CFIR-informed mapping for interpretation. This approach allowed the analysis to remain data-driven while using CFIR to structure explanations of determinants.

We expanded Table 1 (CFIR Construct Mapping for Qualitative Coding) to explicitly link each theme and subtheme to specific CFIR domains and constructs (e.g., Inner Setting: Available Resources; Outer Setting: Patient Needs & Resources; Characteristics of Individuals: Knowledge & Beliefs).

We added illustrative quotations to strengthen the empirical grounding of each theme.

We strengthened the joint display (Table 8) to more directly connect qualitative themes (e.g., stockouts, diagnostic uncertainty, limited supervision) to the quantitative moderators (participant responsiveness, facilitation strategies, intervention complexity) and fidelity outcomes.

Manuscript location:

• Methods → Qualitative analysis

• Table 1: CFIR Construct Mapping for Qualitative Coding

---

## [Editor Report · Decision Letter 1]

12 Apr 2026

Implementation Fidelity of Ethiopia’s Malaria Test-and-Treat Guideline Amid a Resurgence in the Amhara Region: A Mixed Methods Study

PONE-D-25-41084R1

Dear Dr. Lake,

We’re pleased to inform you that your manuscript has been judged scientifically suitable for publication and will be formally accepted for publication once it meets all outstanding technical requirements.

An invoice will be generated when your article is formally accepted. Please note, if your institution has a publishing partnership with PLOS and your article meets the relevant criteria, all or part of your publication costs will be covered. Please make sure your user information is up-to-date by logging into Editorial Manager at Editorial Manager® and clicking the ‘Update My Information' link at the top of the page. For questions related to billing, please contact  and clicking the ‘Update My Information' link at the top of the page. For questions related to billing, please contact billing support..

Kind regards,

Mohammed Hasen Badeso, Epidemiologist

Academic Editor

PLOS One
---

## [Editor Report · Acceptance letter]

PONE-D-25-41084R1

PLOS One

Dear Dr. Lake,

I'm pleased to inform you that your manuscript has been deemed suitable for publication in PLOS One. Congratulations! Your manuscript is now being handed over to our production team.

Kind regards,

on behalf of

Mr Mohammed Hasen Badeso

Academic Editor

PLOS One